# Comparison of Disease Severity Classifications of Chronic Obstructive Pulmonary Disease: GOLD vs. STAR in Clinical Practice

**DOI:** 10.3390/diagnostics14060646

**Published:** 2024-03-19

**Authors:** Koichi Nishimura, Masaaki Kusunose, Ayumi Shibayama, Kazuhito Nakayasu

**Affiliations:** 1Visiting Researcher, National Center for Geriatrics and Gerontology, 7-430, Morioka-cho, Obu 474-8511, Japan; 2Clinic Nishimura, 4-3 Kohigashi, Kuri-cho, Ayabe 623-0222, Japan; 3Department of Respiratory Medicine, National Center for Geriatrics and Gerontology, 7-430, Morioka-cho, Obu 474-8511, Japan; kusunose@ncgg.go.jp; 4Department of Nursing, National Center for Geriatrics and Gerontology, 7-430, Morioka-cho, Obu 474-8511, Japan; ayuminarita3@gmail.com; 5Data Research Section, Kondo P.P. Inc., 17-25, Shimizudani-cho, Tennoujiku, Osaka 543-0011, Japan; nakayasu@mydo-kond.co.jp

**Keywords:** chronic obstructive pulmonary disease (COPD), disease severity, Global Initiative for Chronic Obstructive Lung Disease (GOLD), STaging of Airflow obstruction by Ratio (STAR), St. George’s Respiratory Questionnaire (SGRQ), COPD Assessment Test (CAT)

## Abstract

Background: In chronic obstructive pulmonary disease (COPD), there are two known classifications for assessing what is called disease severity. One is the Global Initiative for Chronic Obstructive Lung Disease (GOLD) classification, which is based on the post-bronchodilator value of FEV_1_ (% reference). The other is the STaging of Airflow obstruction by Ratio (STAR), with four grades of severity in subjects with an FEV_1_/FVC ratio <0.70: STAR 1 ≥0.60 to <0.70, STAR 2 ≥0.50 to <0.60, STAR 3 ≥0.40 to <0.50, and STAR 4 <0.40. Purpose: The aim of this study was to compare the staging of COPD using the GOLD and STAR classifications in clinical practice. Methods: We reanalyzed data from our outpatient cohort study, which included 141 participants with COPD from 2015 to 2023. We compared mortality and COPD-specific health status between the GOLD 1 to 4 groups and the STAR 1 to 4 groups. Results: By simple calculation, GOLD and STAR severity classes coincided in 75 participants (53.2%). The weighted Bangdiwala B value with linear weights was 0.775. The participants were observed for up to 95 months, with a median of 54 months. Death was confirmed in 29 participants (20.5%). In univariate Cox proportional hazards analyses, there was a significant difference in mortality between the GOLD 1 and GOLD 3 + 4 groups, with the GOLD 1 group used as the reference [hazard ratio 4.222 (95% CI 1.298–13.733), *p* = 0.017]. However, there was no statistically significant predictive relationship between STAR 1 and STAR 2, or between STAR 1 and STAR 3 + 4. St. George’s Respiratory Questionnaire (SGRQ) Total and COPD Assessment Test (CAT) scores were significantly different between all GOLD groups, except for the CAT score between GOLD 1 and GOLD 2. The SGRQ Total and CAT scores were significantly different between STAR 1 and STAR 3 + 4, but not between STAR 1 and STAR 2. Conclusion: From the perspective of all-cause mortality and COPD-specific health status, the GOLD classification is more discriminative than STAR.

## 1. Introduction

There is no doubt that airflow limitation is the definition of chronic obstructive pulmonary disease (COPD), but how to define the severity of COPD has historically been the subject of debate among physicians and researchers around the world. Despite being the third leading cause of death worldwide and the large number of sufferers, it may come as a surprise that defining the severity of COPD is not an easy task although efforts such as the Global Initiative for Chronic Obstructive Lung Disease (GOLD) have sought to address this [1,2]. The concept that a diagnosis of COPD could be made by assessing the ratio of forced expiratory volume in one second (FEV_1_) to forced vital capacity (FVC) and the severity of COPD by the ratio of FEV_1_ to predicted FEV_1_ became mainstream in the 1990s and has been incorporated into the GOLD document since its inception [1]. However, since the start of the 21st century, several studies have reported that dyspnea, exercise tolerance or physical activity are better predictors of outcome than FEV_1_ [3,4,5], and an increasing number of reports have emphasized the importance of patient-reported outcomes and acute exacerbations [6,7,8,9]. However, the current version of the GOLD document states that, in the presence of an FEV_1_/FVC ratio <0.7, the assessment of airflow limitation severity in COPD (note that this may be different from severity of the disease) is based on the post-bronchodilator value of FEV_1_ (% reference), or FEV_1_% predicted (ppFEV_1_) [2].

Furthermore, there remains the larger question of what factors should determine the severity of COPD, as there may be disagreement on how to define disease severity in general. For example, is disease severity strongly associated with mortality prediction or is severity the fact that patients suffer more from COPD? Many severity classifications proposed as composite markers have been based on their superiority as mortality predictors [10,11,12], but if the priority is to evaluate the fact that patients are suffering more, then the inclusion of quality of life and related indicators such as health status would be more precise [13].

There have been reports that alternative methods of severity classification may be preferable to the percentage-based classification method for the predicted value of FEV_1_, as in GOLD stages 1 to 4 [14,15,16,17]. A recent, thought-provoking report has suggested that one particular method may be superior to the GOLD criteria as a severity classification for COPD [18]. The classification in question is named STAR (STaging of Airflow obstruction by Ratio), identified four grades of severity in subjects with an FEV1/FVC ratio <0.70: STAR 1 ≥0.60 to <0.70, STAR 2 ≥0.50 to <0.60, STAR 3 ≥0.40 to <0.50, and STAR 4 <0.40 and it has sparked a heated debate [19,20,21,22,23]. However, the concerns about the STAR have remained theoretical or conceptual. Since the authors are in the process of conducting a cohort study summary of our own institution [24,25], we attempted to compare STAR and GOLD using data obtained from our clinical practice since it is important to examine whether their hypotheses work for populations with very different backgrounds. The aim of the present study is to compare the staging of COPD using the GOLD and STAR classifications in clinical practice in Japan.

## 2. Materials and Methods

From 2013 to 2023, the Respiratory Medicine Outpatient Clinic at the National Center for Geriatrics and Gerontology collected data for a longitudinal study on individuals diagnosed with COPD [24,25]. This study included 141 patients aged 50 and above who had smoked for at least 10 pack-years, had a post-bronchodilator FEV_1_/FVC ratio below 0.7, had no abnormal chest X-ray findings, had no active pulmonary diseases or unmanaged comorbid conditions, and had not made any changes to their treatment plans in the preceding four weeks. Patients with a history of asthma or recent COPD exacerbations within the preceding three months were excluded. All participants had received at least six months of outpatient care prior to this study to ensure stability before any new interventions were introduced. This study was approved by the Ethics Committee of the National Center for Geriatrics and Gerontology (No. 1138-3) (updated on 12 July 2020) and adhered to the principles of the Declaration of Helsinki. Written informed consent was obtained from all participants. Enrolled participants who met the study criteria and provided informed consent underwent biannual evaluations, which included lung function tests after bronchodilator use and assessments of their health status. The baseline for this analysis was established in 2015, when the Kihon Checklist was incorporated into this study to help identify frailty [26]. This paper revisits and reanalyzes the data of the same cohort previously discussed in a prior report, with continuous recruitment of participants from February 2015 to February 2022 [25]. All participants were followed and evaluated for a maximum of six and a half years until January 2023. The time from enrolment to the last involvement or event was documented for examination.

Participants were instructed to abstain from using bronchodilators for at least 12 h before visiting the research facility. Spirometry was performed more than one hour after the administration of a dry powder, long-acting bronchodilator, supervised by a physician, using a CHESTAC-8800 spirometer (Chest, Tokyo, Japan). The highest values from three attempts were recorded, and residual volume (RV) was determined using the closed-circuit helium technique. All procedures were conducted by trained lab technicians in accordance with the guidelines of the American Thoracic Society and the European Respiratory Society [27].

Using FEV_1_ and FVC measured at baseline, patients were classified into four groups each by two methods, the GOLD classification and the STAR classification. According to the former, i.e., the GOLD classification of airflow limitation, patients with FEV_1_ ≥ 80% predicted were classified as GOLD 1 patients, patients with 50% ≤ FEV_1_ < 80% predicted as GOLD 2, patients with 30% ≤ FEV_1_ < 50% predicted were classified as GOLD 3, and patients with FEV_1_ < 30% predicted were classified as GOLD 4 [1,2]. The reference values for lung function were provided by the Japanese Respiratory Society [28]. According to the latter, i.e., STAR classification, patients with 0.7 > FEV_1_/FVC ≥ 0.60 were classified as the STAR 1 patient group, patients with 0.6 > FEV_1_/FVC ≥ 0.5 as STAR 2, patients with 0.5 > FEV_1_/FVC ≥ 0.4 as STAR 3, and patients with 0.4 > FEV_1_/FVC as STAR 4 [18].

Japanese versions of St. George’s Respiratory Questionnaire (SGRQ) (version 2) and the COPD Assessment Test (CAT) were used to assess COPD-specific health status [29,30,31,32]. The SGRQ consists of 50 items and three domains (Symptoms, Activity, and Impact). For this analysis, the focus was on the overall score, which ranges from 0 to 100, with higher scores indicating poorer health [29]. The CAT score ranges from 0 to 40, with higher scores indicating more severe impairment [31]. We noted the time between the start of this study and the last follow-up or event. To confirm the survival status of the participants, we either contacted them directly or reached out to their families or healthcare providers if they were not attending follow-up appointments. The findings are presented as mean ± standard deviation. Statistical significance was determined using a *p*-value of less than 0.05. We used Bangdiwala plots to descriptively examine the concordance of the two severity classification systems. We also quantified the agreement using the linearly weighted Bangdiwala’s B value. To identify differences among groups, we used Steel–Dwass and Fisher’s exact tests with Bonferroni correction. We compared the primary endpoint, all-cause mortality, between classification methods. We explored the associations between various measurements and mortality using univariate Cox proportional hazards models. The results are expressed as hazard ratios (HR) and 95% confidence intervals (CI). We used the C-index to compare the predictive capabilities of different models, where values closer to 1.0 indicate superior risk prediction. Furthermore, we analyzed time-to-event data using Kaplan–Meier curves and log-rank tests with Bonferroni correction.

## 3. Results

### 3.1. Patient Characteristics

A study was conducted on a sample of 141 individuals (130 males) with varying degrees of COPD, ranging from mild to very severe (Table 1). The participants had a mean age of 75.2 ± 6.7 years, and their FEV_1_ values were 1.74 ± 0.54 L (69.8 ± 20.1% predicted). Based on the GOLD classification of airflow limitation, 43 individuals (30.5%) were classified as GOLD 1, 74 (52.5%) as GOLD 2, 19 (13.5%) as GOLD 3, and 5 (3.5%) as GOLD 4 (Table 2). According to the STAR, 64 individuals (45.4%) were classified as STAR 1, 39 (27.7%) as STAR 2, 24 (17.0%) as STAR 3, and 14 (9.9%) as STAR 4. Due to the relatively small number of patients classified as GOLD 3 and GOLD 4, as well as STAR 3 and STAR 4, it was decided to combine the 24 patients classified as GOLD 3 and 4 into a single group. Likewise, the 38 patients classified as STAR 3 and 4 were combined into another group for subsequent analyses.

### 3.2. Concordance between GOLD and STAR

The distribution of participants according to disease severity, as classified by GOLD and STAR, is presented in Table 3. A comparison of the GOLD and STAR stage classifications shows agreement in 75 out of 141 subjects (53.2%) (Table 3). The agreement between GOLD and STAR severity classes was further evaluated using Bangdiwala agreement charts, as depicted in Figure 1. Black squares and a 45-degree diagonal line touching the edges of each square represent perfect agreement. The shades of the rectangles indicate the level of agreement, with darker rectangles indicating complete agreement and progressively lighter shades indicating partial agreement. When only perfect matches were considered, Bangdiwala’s B value was 0.341. However, when misalignments were weighted based on their magnitude, the agreement (weighted Bangdiwala’s B value with linear weights) between GOLD and STAR severity classes was found to be 0.775.

### 3.3. COPD-Specific Health Status

The SGRQ Total and CAT scores were compared using the Steel–Dwass test between the three groups classified by GOLD and the three groups classified by STAR (Table 2). The SGRQ Total scores showed significant differences between all GOLD groups (GOLD 1 vs. GOLD 2, *p* = 0.011, GOLD 1 vs. GOLD 3 + 4 and GOLD 2 vs. GOLD 3 + 4, both *p* < 0.001). The CAT scores also showed significant differences between GOLD groups (GOLD 1 vs. GOLD 3 + 4 and GOLD 2 vs. GOLD 3 + 4, both *p* < 0.001) except for the CAT score between GOLD 1 and GOLD 2 (*p* = 0.250). On the other hand, the SGRQ Total and CAT scores were significantly different between STAR 1 and STAR 3 + 4 (SGRQ Total, *p* < 0.001 and CAT, *p* = 0.002), but not between STAR 1 and STAR 2 (SGRQ Total, *p* = 0.945 and CAT, *p* = 0.612). In the comparison between STAR 2 and STAR 3 + 4, the SGRQ Total score was significantly different (*p* = 0.004), but the CAT score was not (*p* = 0.081). From a health status perspective, the GOLD classification was found to be more discriminative than the STAR classification.

### 3.4. Survival

The participants were observed for up to 95 months, with an average observation period of 54.5 (±27.4) months and a median of 54 months. Among the 141 study participants, 29 (20.5%) were confirmed to have died. To examine the associations between the severity classification of GOLD or STAR and all-cause mortality, univariate Cox proportional hazards analyses were conducted. A significant difference was found in mortality between the GOLD 1 and GOLD 3 + 4 groups, using the GOLD 1 group as the reference [hazard ratio 4.222 (95% CI 1.298–13.733), *p* = 0.017]. There was no significant difference between the GOLD 1 and GOLD 2 groups [HR 2.658 (95% CI 0.887–7.9619), *p* = 0.081]. However, there was no statistically significant predictive relationship observed between the STAR 1 and STAR 2 groups, nor between the STAR 1 and STAR 3 + 4 groups [HR 1.543 (95% CI 0.626–3.800), *p* = 0.346 and HR 1.791 (95% CI 0.744–4.313), *p* = 0.194, respectively]. The C-index for GOLD was 0.637, which was higher than the C-index of 0.578 for STAR. Figure 2 and Figure 3 depict the survival curves for disease severity classes based on GOLD and STAR stages, respectively. To compare GOLD 1 and GOLD 2, GOLD 1 and GOLD 3 + 4, and GOLD 2 and GOLD 3 + 4, log-rank tests with Bonferroni corrections were applied. A significant difference was found between GOLD 1 and GOLD 3 + 4 (*p* = 0.024), but not between GOLD 1 and GOLD 2 (*p* = 0.224) or between GOLD 2 and GOLD 3 + 4 (*p* = 0.843). The log-rank tests with Bonferroni corrections were also used to analyze the differences between STAR 1, STAR 2, and STAR 3 + 4, but none of these comparisons were found to be significant.

## 4. Discussion

The present study compared the staging of COPD using the GOLD and STAR classifications and found that the GOLD classification is a better predictor of mortality than STAR. It also found that the GOLD classification outperforms the STAR classification in assessing COPD-specific health status measurements, such as the SGRQ Total or CAT score. From these two perspectives, GOLD was considered superior to STAR as an assessment of COPD severity in routine clinical settings. This finding is surprising considering a recent study by Bhatt et al. that showed the STAR classification to be superior to GOLD staging. However, it is important to note that the populations studied in these two studies differed markedly. Bhatt et al.’s study included 12,000 participants from the COPDGene study and two additional large studies [18], while ours focused on patients from a medium-sized hospital in Japan.

The question of whether the priority in COPD severity classification should be placed on epidemiological functionality or usefulness for treating patients requires thorough debate. One advantage of the STAR classification over GOLD is its ability to differentiate stage 1 from the absence of airflow obstruction, as shown in Bhatt et al.’s study [18]. This distinction may be more important from an epidemiological perspective and may hold less importance for physicians treating patients with COPD. However, the results of our study do not support the claim that STAR is better than GOLD at distinguishing between healthy subjects and patients with mild COPD as we did not include healthy subjects in our study. Respiratory physicians treating COPD patients typically encounter more severe cases, so there may be little benefit in using the STAR classification instead of the GOLD classification. In our study, we examined both GOLD and STAR classifications as indicators of disease severity. Although we thought that predictors of mortality and COPD-specific health status played an important role in these classifications, Bhatt et al. took a more comprehensive approach to validate the severity classification by exploring associations with COPD symptoms, exercise capacity, lung disease using computed tomography, exacerbations, FEV_1_ change, and physiological factors [18].

This study compared the GOLD and STAR severity classifications based on the hypothesis that COPD severity is expressed by two concepts: mortality prediction and COPD-specific health status, which is most relevant to how patients are suffering from COPD. Both analyses of mortality prediction and health status showed that the GOLD classification was superior to the STAR in terms of its ability to discriminate. Despite the different analyses, many studies have shown that the SGRQ and CAT scores are significant mortality predictors [24]. Therefore, as determinants of COPD severity, mortality prediction and health status may have some similarities in terms of patient categorization although they are assessed in different ways.

The important issue here concerns the scientific meaning of ‘seriously ill’. Clinicians often claim that a patient is sick or in poor health without sufficient evidence to support their statements and this is a problem that needs to be addressed. Initially, the GOLD document defined severity as a disease classification, but it has since been updated to clarify that it refers specifically to the severity of airflow limitation. Other composite markers, such as the BODE and ADO index, as well as multidimensional staging systems, have also been proposed as classifications of COPD severity [10,11]. However, most of these markers have only been validated as predictors of mortality. The determination of severity classification for diseases other than COPD varies. While severity scales exist and have been used for many diseases, there is no consensus on how to determine severity. These scales range from those that account for diseases with low mortality rates to those with high mortality rates. However, most of these scales use a four-point system, and there are no standardized guidelines for creating severity scales. According to Wikipedia, Severity of Illness (SOI) is defined as the extent of organ system derangement or physiological decompensation a patient experiences [33]. The Agency for Healthcare Research and Quality (AHRQ) website explains that in disease staging, severity refers to the likelihood of death or organ failure resulting from disease progression, independent of the treatment process [34,35]. It is important to note that these general descriptions of disease stage are influenced by factors such as hospital administration and health economics, and are not directly applicable to determining the stage of COPD.

The current study has several limitations primarily due to its design. A major limitation of our study is the small number of cases. With a larger sample size, there may be significant differences among all severity groups, whether using GOLD or STAR. However, conducting such a study with a large group of patients in a single medical facility is extremely rare. Therefore, in clinical practice, it should be acknowledged that the GOLD classification is superior to the STAR classification as a predictor of mortality and COPD-specific health status. Furthermore, this study was conducted at a single center, which means that the scope was limited to the cohort of COPD patients treated at our facility. While it is a comprehensive study within this hospital during the research period, its findings might not be broadly applicable. It also differs slightly from previous research in this area as it included a greater number of older patients and a relatively smaller number of participants with severe and very severe COPD. Additionally, the predominance of male participants in our study restricts the generalizability of our results to the female COPD population. This gender disparity reflects the lower prevalence of diagnosed COPD among women in Japan accurately representing the clinical landscape of COPD in our demographic [36]. A selection bias may be present as this study only includes patients who were able to consistently visit our outpatient clinic. As a result, we may have overlooked a segment of the COPD population, particularly asymptomatic individuals who are unaware of their condition or those unable to attend regular appointments due to severe physical constraints.

## 5. Conclusions

From the perspective of all-cause mortality and COPD-specific health status, the GOLD classification is more discriminative than STAR. In clinical practice, the results indicate that GOLD may be a superior method of classifying COPD severity than STAR. However, a general discussion should be had to identify the specific factors that should determine the severity classification.

## Figures and Tables

**Figure 1 diagnostics-14-00646-f001:**
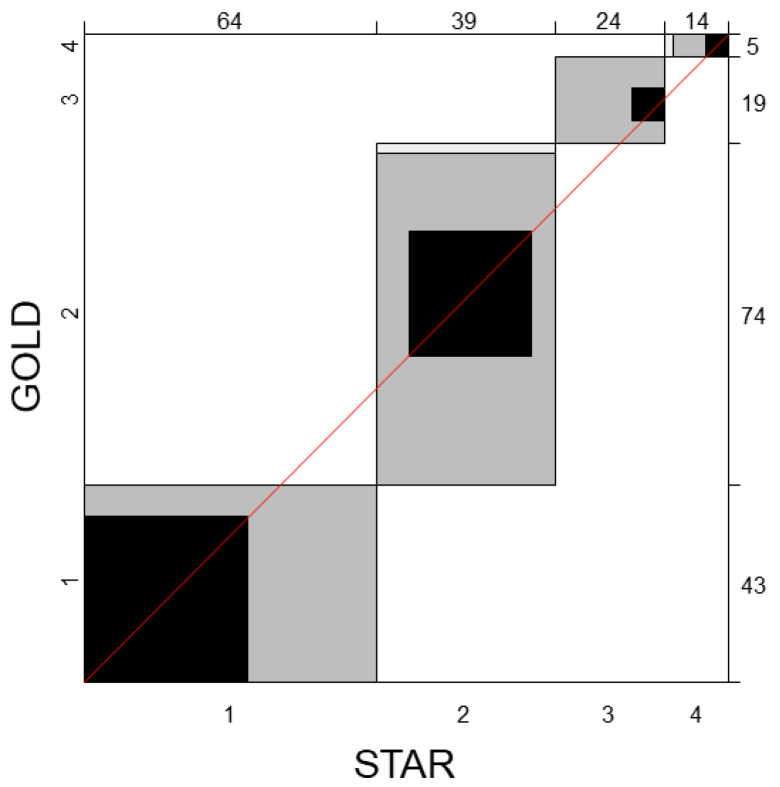
Bangdiwala agreement charts comparing the severity classification of airflow obstruction in 141 patients with COPD using two different schemes: the Global Initiative for Chronic Obstructive Lung Disease (GOLD) and the STaging of Airflow obstruction by Ratio (STAR) severity scheme.

**Figure 2 diagnostics-14-00646-f002:**
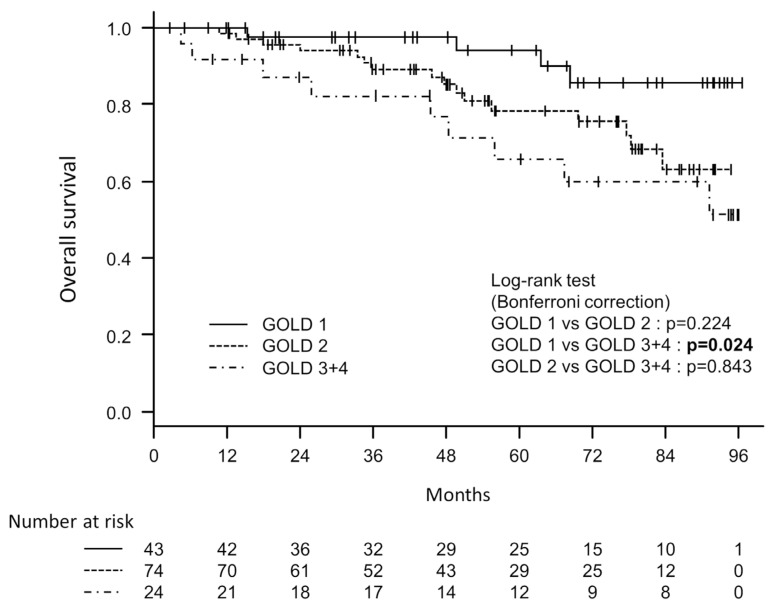
Kaplan–Meier survival curves based on three groups (GOLD 1, GOLD 2 and GOLD 3 + 4) classified using the Global Initiative for Chronic Obstructive Lung Disease (GOLD) document.

**Figure 3 diagnostics-14-00646-f003:**
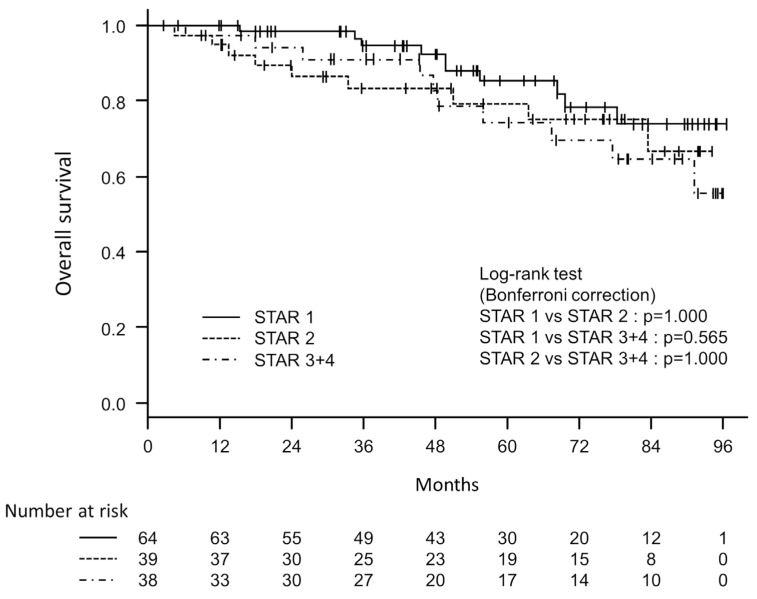
Kaplan–Meier survival curves based on three groups (STAR 1, STAR 2 and STAR 3 + 4) classified using the STaging of Airflow obstruction by Ratio (STAR) severity scheme.

**Table 1 diagnostics-14-00646-t001:** Patient characteristics at baseline in 141 subjects with COPD.

		Mean ± SD
Age	years	75.2 ± 6.7
BMI	kg/m^2^	22.8 ± 3.3
Cumulative smoking	pack-years	59.1 ± 32.0
FEV_1_	Liters	1.74 ± 0.54
FEV_1_	%pred	69.8 ± 20.1
FEV_1_/FVC	%	56.0 ± 10.7
RV	%pred	124.6 ± 52.9
RV/TLC	%	45.0 ± 11.2
DLco ^(1)^	mL/min/mmHg	11.99 ± 5.04
PaO_2_ ^(2)^	mmHg	79.2 ± 9.0
SGRQ total score	(0–100)	22.6 ± 16.4
CAT score	(0–40)	8.6 ± 7.0
Sex	male/female	130/11
GOLD stage	GOLD 1/GOLD 2/GOLD 3 + 4	43/74/24 (30.5%/52.5%/17.0%)
STAR stage	STAR 1/STAR 2/STAR 3 + 4	64/39/38 (45.4%/27.7%/27.0%)

^(1)^ n = 140, ^(2)^ one patient receiving oxygen. Numbers in parentheses denote possible score range. SGRQ, St. George’s Respiratory Questionnaire; CAT, the COPD Assessment Test; GOLD, Global Initiative for Chronic Obstructive Lung Disease; STAR, STaging of Airflow obstruction by Ratio.

**Table 2 diagnostics-14-00646-t002:** Comparison of clinical and physiological backgrounds and COPD-specific health status in the classification of GOLD 1 to 4 and STAR 1 to 4.

		GOLD 1 N = 43 (30.5%)		GOLD 2 N = 74 (52.5%)		GOLD 3 + 4 N = 24 (17.0%)		comparison between groups (*p*-value)		
		n	mean ± SD	n	mean ± SD	n	mean ± SD	GOLD 1 vs. 2	GOLD 1 vs. 3 + 4	GOLD 2 vs. 3 + 4
Age	years	43	74.6 ± 5.9	74	75.4 ± 7.4	24	75.7 ± 5.9	0.735 ‡	0.953 ‡	0.989 ‡
BMI	kg/m^2^	43	23.4 ± 2.6	74	22.9 ± 3.7	24	21.6 ± 3.0	0.584 ‡	0.046 ‡	0.261 ‡
TLC	%pred	43	107.3 ± 16.8	74	104.8 ± 29.2	24	103.9 ± 22.5	0.575 ‡	0.443 ‡	0.985 ‡
RV	%pred	43	103.9 ± 25.5	74	129.3 ± 59.0	24	147.4 ± 58.2	0.024 ‡	<0.001 ‡	0.111 ‡
RV/TLC	%	43	37.3 ± 5.2	74	46.7 ± 11.8	24	53.4 ± 9.1	<0.001 ‡	<0.001 ‡	0.003 ‡
DLco	mL/min/mmHg	43	13.59 ± 4.00	74	11.53 ± 4.69	23	10.46 ± 6.99	0.016 ‡	0.012 ‡	0.290 ‡
Sex	male/female	37 (26.2%)/6 (4.3%)		69 (48.9%)/5 (3.5%)		24 (17.0%)/0 (0%)		0.627 †	0.242 †	0.990 †
SGRQ total score	(0–100)	43	13.9 ± 9.5	74	22.2 ± 15.5	24	39.4 ± 16.8	0.011 ‡	<0.001 ‡	<0.001 ‡
CAT score	(0–40)	43	6.0 ± 5.1	74	8.0 ± 6.4	24	15.1 ± 7.8	0.250 ‡	<0.001 ‡	<0.001 ‡
		STAR 1 N = 64 (45.4%)		STAR 2 N = 39 (27.7%)		STAR 3 + 4 N = 38 (27.0%)		comparison between groups (*p*-value)		
		n	mean ± SD	n	mean ± SD	n	mean ± SD	STAR 1 vs. 2	STAR 1 vs. 3 + 4	STAR 2 vs. 3 + 4
Age	years	64	76.0 ± 6.2	39	74.9 ± 7.5	38	74.1 ± 6.8	0.730 ‡	0.152 ‡	0.612 ‡
BMI	kg/m^2^	64	23.4 ± 3.3	39	22.9 ± 3.4	38	21.8 ± 2.9	0.326 ‡	0.035 ‡	0.401 ‡
TLC	%pred	64	101.5 ± 24.7	39	107.6 ± 27.1	38	109.8 ± 21.8	0.449 ‡	0.052 ‡	0.617 ‡
RV	%pred	64	110.1 ± 47.6	39	134.7 ± 61.9	38	138.8 ± 45.9	0.020 ‡	<0.001 ‡	0.222 ‡
RV/TLC	%	64	42.0 ± 10.0	39	47.6 ± 13.9	38	47.3 ± 9.0	0.046 ‡	<0.001 ‡	0.565 ‡
DLco	mL/min/mmHg	64	13.27 ± 4.49	39	12.32 ± 5.54	37	9.42 ± 4.54	0.274 ‡	<0.001 ‡	0.053 ‡
Sex	male/female	56 (39.7%)/8 (5.7%)		37 (26.2%)/2 (1.4%)		37 (26.2%)/1 (0.7%)		0.936 †	0.445 †	1.000 †
SGRQ total score	(0–100)	64	19.1 ± 14.8	39	19.8 ± 14.4	38	31.4 ± 18.0	0.945 ‡	<0.001 ‡	0.004 ‡
CAT score	(0–40)	64	6.7 ± 5.4	39	8.3 ± 6.9	38	12.1 ± 8.2	0.612 ‡	0.002 ‡	0.081 ‡

‡: Steel–Dwass test; †: Fisher’s exact test with Bonferroni correction. Numbers in parentheses denote possible score range. GOLD, Global Initiative for Chronic Obstructive Lung Disease; STAR, STaging of Airflow obstruction by Ratio; SGRQ, St. George’s Respiratory Questionnaire; CAT, the COPD Assessment Test.

**Table 3 diagnostics-14-00646-t003:** Distribution of the number in the classification of GOLD (Global Initiative for Chronic Obstructive Lung Disease) 1 to 4 and STAR (STaging of Airflow obstruction by Ratio) 1 to 4.

	STAR 1	STAR 2	STAR 3	STAR 4
GOLD 1	36 (25.3%)	7 (5.0%)	0 (0%)	0 (0%)
GOLD 2	28 (19.9%)	27 (19.1%)	17 (12.1%)	2 (1.4%)
GOLD 3	0 (0%)	5 (3.5%)	7 (5.0%)	7 (5.0%)
GOLD 4	0 (0%)	0 (0%)	0 (0%)	5 (3.5%)

## Data Availability

Anonymized participant data will be made available upon reasonable request to the corresponding author.

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
