# Peer review of "Comparison of Disease Severity Classifications of Chronic Obstructive Pulmonary Disease: GOLD vs. STAR in Clinical Practice"

_diagnostics, 2024, doi:10.3390/diagnostics14060646_

Round 1

Reviewer 1 Report

Comments and Suggestions for Authors

Abstract:

The aim of the study is not clear

Introduction:

The authors did not clearly mention the aim of the work. What is written in 76-78 is not clear objectives, it is just a hypothesis. Also, for reference 24 and the sentence related (lines 75-76), is not related to the introduction and should be remove.

Methods:

The authors should mention their study type clearly. It is a retrospective cohort study as they re-analyzed previously studied group. Please add 

The methodology of the study is deficient. The authors did not mention any data about the patient classification, nothing is written about disease severity classification neither GOLD and STAR stratification. The severity scales should be well described in the methods and supported by references.

Results:

The first paragraph of the results: the authors 1st mentioned the diseases severity according to GOLD and STAR which should be the methods (the description not the number of patients in each group). Please correct

Also in the same paragraph, lines 142-144 are not related to results. This part should be present in the discussion and study limitations. Please correct

In table 1: please add the percentage for the categories.

Lines 159-162 should be in the 1st paragraph of the results as they are presented in table 1 and it was not clear. Please correct

There is nothing in text about tables 2 and 3. The authors should highlight the most important points in these table in the text under the subtitle 3.2 Concordance between GOLD and STARAlso, all the number presented for the categorized data in table 3 and 2 should be supported by percentage. So, please correct the missing data.

Table 3 should be before table 2 as it is describing figure 1. Please correct

On figure 2 and 3, add the log rank and p value.

In the last paragraph of the results under the subtitle 3.4 COPD-specific health statusthe authors should write the p value in the text with their description with referral to table 2. Also, this part should be presented before the part of the survival. Please correct

Lines 229-231 should be in the discussion not in the results. Please remove from here.

Discussion:

Lines 243-249 belong to the limitation section of the study, not here. Please correct.

The discussion is to some extent deficient. There is a repetition of the same sentences in different writing throughout the discussion. I suggest the authors to organize it better and concentrate on better references in their comparison. There are many references in the literature highlighted CAT score as the best disease-specific heath status assessment and high relation to mortality rather than Wikipedia for example.   

Also, the limitations of the study, despite being well described, it is missing references as that regarding the prevalence of COPD in males more than females in Japan. Please correct

Conclusion: It is ok and relevant

References: to be reviewed 

Comments on the Quality of English Language

The language is ok, but there are some sentences not so clear especially in the discussion. I suggest the authors to review the English language with a native English speaker

Author Response

Author's Reply to the Review Report (Reviewer 1)

  1. Abstract: The aim of the study is not clear

Response: Thank you very much for your concern. We have added the following sentence to the abstract in the revision; “Purpose: The aim of this study was to compare the staging of COPD using the GOLD and STAR classifications in clinical practice.” (lines 19 to 21 in the revision)

  1. Introduction: The authors did not clearly mention the aim of the work. What is written in 76-78 is not clear objectives, it is just a hypothesis.

Response: We appreciate your comment. We have inserted the following sentence at the end of the introduction section in the revision; “The aim of the present study is to compare the staging of COPD using the GOLD and STAR classifications in clinical practice in Japan.” (lines 77 to 78 in the revision)

  1. Also, for reference 24 and the sentence related (lines 75-76), is not related to the introduction and should be remove.

Response: Thank you very much for your effort and patience in reviewing our paper and providing advice. According to your advice, the following sentence has been removed from the revision: “Although Bhatt and colleagues reported the validation of the STAR in a large population sample of over 12,000 participants including the COPDGene (Genetic Epidemiology of COPD) study [17], our study population is only a small one with fewer than 150 patients [24].” (lines 73 to 76 in the original manuscript)

  1. The methodology of the study is deficient. The authors did not mention any data about the patient classification, nothing is written about disease severity classification neither GOLD and STAR stratification. The severity scales should be well described in the methods and supported by references.

Response: According to your advice, we have added the following paragraph to the Methods section in the revision with references that accounted for; “Using FEV1 and FVC measured at baseline, patients were classified into four groups each by two methods, the GOLD classification, and the STAR classification. According to the former, i.e., GOLD classification of airflow limitation, patients with FEV1 ≥ 80% predicted were classified as GOLD 1 patients, patients with 50% ≤ FEV1 < 80% predicted as GOLD 2, patients with 30% ≤ FEV1 < 50% predicted were classified as GOLD 3, and patients with FEV1 < 30% predicted were classified as GOLD 4 [1,2]. The reference values for lung function were provided by the Japanese Respiratory Society [28]. According to the latter, i.e. STAR classification, patients with 0.7 > FEV1/FVC ≥ 0.60 were classified as the STAR 1 patient group, patients with 0.6 > FEV1/FVC ≥ 0.5 as STAR 2, patients with 0.5 > FEV1/FVC ≥ 0.4 as STAR 3, and patients with 0.4 > FEV1/FVC as STAR 4 [18].” (lines 110 to 119 in the revision)

  1. Results: The first paragraph of the results: the authors 1st mentioned the diseases severity according to GOLD and STAR which should be the methods (the description not the number of patients in each group). Please correct.

Response: According to your advice, the description of patient severity has been moved to the Methods section in the revision. Only the number of people in each group is described here.

  1. Also in the same paragraph, lines 142-144 are not related to results. This part should be present in the discussion and study limitations. Please correct.

Response: We are grateful to you for your helpful advice. This sentence has been moved from the Results section in the original manuscript (lines 141 to 144 in the original manuscript) to the paragraph on study limitations in the discussion in the revised manuscript (lines 298 to 301 in the revision). It now reads as follows, “It also differs slightly from previous research in this area as it included a greater number of older patients and a relatively smaller number of participants with severe and very severe COPD.”

  1. In table 1: please add the percentage for the categories.

Response: According to your advice, the percentage for the categories has been added to Table 1 of the revision.

Table 1. Patient characteristics at baseline in 141 subjects with COPD

mean

SD

Age

years

75.2

±

6.7

BMI

kg/m2

22.8

±

3.3

Cumulative Smoking

pack-years

59.1

±

32.0

FEV1

 Liters

1.74

±

0.54

FEV1

%pred

69.8

±

20.1

FEV1/FVC

%

56.0

±

10.7

RV

%pred

124.6

±

52.9

RV/TLC

%

45.0

±

11.2

DLco1)

mL/min/mmHg

11.99

±

5.04

PaO22)

mmHg

79.2

±

9.0

SGRQ Total score

(0 - 100)

22.6

±

16.4

CAT score

(0 - 40)

8.6

±

7.0

Sex

male / female

130 / 11

GOLD stage

GOLD 1/GOLD 2/GOLD 3+4

43 / 74 / 24

(30.5%/52.5%/17.0%)

STAR stage

STAR 1/STAR 2/STAR 3+4

64 / 39 / 38

(45.4%/27.7%/27.0%)

1) n=140, 2) one patient receiving oxygen. Numbers in parentheses denote possible score range. SGRQ, St. George’s Respiratory Questionnaire; CAT, the COPD Assessment Test; GOLD, Global Initiative for Chronic Obstructive Lung Disease; STAR, STaging of Airflow obstruction by Ratio.

  1. Lines 159-162 should be in the 1st paragraph of the results as they are presented in table 1 and it was not clear. Please correct.

Response: We appreciate your comment. These sentences have been moved to the first paragraph in the revision (3.1. Patient characteristics) (lines 149 to 153 in the revision).

  1. There is nothing in text about tables 2 and 3. The authors should highlight the most important points in these table in the text under the subtitle 3.2 Concordance between GOLD and STAR. Also, all the number presented for the categorized data in table 3 and 2 should be supported by percentage. So, please correct the missing data.

Response: We are grateful to you for your helpful advice, indicating that confusion was caused by Tables 2 and 3 having no associated text. The Revision adds a reference to Table 2 in one place in the first paragraph (3.1. Patient characteristics), a reference to Table 3 in one place in the second paragraph (3.2. Concordance between GOLD and STAR) and a reference to Table 2 in the third paragraph (3.3. COPD-specific health status). The results for Table 2 are presented in the text in two separate paragraphs. We assume that the omission of the reference in these tables has been taken as an omission in the description of the results in relation to the tables indicated. Furthermore, in each of the two tables, the number of patients is indicated with a percentage.

Table 3. Distribution of the number in the classification of GOLD (Global Initiative for Chronic Obstructive Lung Disease) 1 to 4 and STAR (STaging of Airflow obstruction by Ratio) 1 to 4

STAR 1

STAR 2

STAR 3

STAR 4

GOLD 1

36 (25.3%)

7 (5.0%)

0 (0%)

0 (0%)

GOLD 2

28 (19.9%)

27 (19.1%)

17 (12.1%)

2 (1.4%)

GOLD 3

0 (0%)

5 (3.5%)

7 (5.0%)

7 (5.0%)

GOLD 4

0 (0%)

0 (0%)

0 (0%)

5 (3.5%)

  1. Table 3 should be before table 2 as it is describing figure 1. Please correct.

Response: The Revision adds a reference to Table 2 in one place in the first paragraph (3.1. Patient characteristics), a reference to Table 3 in one place in the second paragraph (3.2. Concordance between GOLD and STAR) and a reference to Table 2 in the third paragraph (3.3. COPD-specific health status). The results for Table 2 are presented in the text in two separate paragraphs. Therefore, the order of Tables 2 and 3 has not been changed from the original manuscript since tables should be inserted into the main text close to their first citation and must be numbered following their number of appearances.

  1. On figure 2 and 3, add the log rank and p value.

Response: We have added the information with p values in Figures 2 and 3 in the revision.

  1. In the last paragraph of the results under the subtitle 3.4 COPD-specific health status, the authors should write the p value in the text with their description with referral to table 2. Also, this part should be presented before the part of the survival. Please correct.

Response: First of all, in the Results section, following your suggestion, the original order of sections, 3.3. Survival and 3.4. COPD-specific health status has been changed to, 3.3. COPD-specific health status, and then 3.4. Survival in the revision.

    In addition, the p-values have been added throughout the paragraph entitled 3.3. COPD-specific health status in the revision. This now reads as follows; “The SGRQ Total and CAT scores were compared using the Steel-Dwass test between the three groups classified by GOLD and the three groups classified by STAR (Table 2). The SGRQ Total scores showed significant differences between all GOLD groups (GOLD 1 vs. GOLD 2, p=0.011, GOLD 1 vs. GOLD 3+4 and GOLD 2 vs. GOLD 3+4, both p<0.001). The CAT scores also showed significant differences between GOLD groups (GOLD 1 vs. GOLD 3+4 and GOLD 2 vs. GOLD 3+4, both p<0.001) except for the CAT score between GOLD 1 and GOLD 2 (p=0.250). On the other hand, the SGRQ Total and CAT scores were significantly different between STAR 1 and STAR 3+4 (SGRQ Total, p<0.001 and CAT, p=0.002), but not between STAR 1 and STAR 2 (SGRQ Total, p=0.945 and CAT, p=0.612). In the comparison between STAR 2 and STAR 3+4, the SGRQ Total score was significantly different (p=0.004), but the CAT score was not (p=0.081). From a health status perspective, the GOLD classification was found to be more discriminative than the STAR classification.” (lines 186 to 198 in the revision).

  1. Lines 229-231 should be in the discussion not in the results. Please remove from here.

Response: We thank you for your comments. We have deleted the following sentence since it should be moved to the discussion section, but we intended to avoid duplication there; “From a health status perspective, the GOLD classification was found to be more discriminative than the STAR classification.” (lines 229-231 in the original manuscript).

  1. Discussion: Lines 243-249 belong to the limitation section of the study, not here. Please correct.

Response: According to your advice, the sentences you mentioned were in the first paragraph of the discussion section in the original manuscript, but in the revision, we have moved them to the paragraph on study limitations (lines 289 to 295 in the revision).

  1. The discussion is to some extent deficient. There is a repetition of the same sentences in different writing throughout the discussion. I suggest the authors to organize it better and concentrate on better references in their comparison. There are many references in the literature highlighted CAT score as the best disease-specific heath status assessment and high relation to mortality rather than Wikipedia for example. 

Response: We thank you for your comments. One paragraph has been inserted in the revision. This reads as follows; “This study compared the GOLD and STAR severity classifications based on the hypothesis that COPD severity is expressed by two concepts: mortality prediction and COPD-specific health status, which is most relevant to how patients are suffering from COPD. Both analyses of mortality prediction and health status showed that the GOLD classification was superior to the STAR in terms of its ability to discriminate. Despite the different analyses, many studies have shown that the SGRQ and CAT scores are significant mortality predictors [24]. Therefore, as determinants of COPD severity, mortality prediction and health status may have some similarities in terms of patient categorization although they are assessed in different ways.” (lines 259 to 267 in the revision).

  1. Also, the limitations of the study, despite being well described, it is missing references as that regarding the prevalence of COPD in males more than females in Japan. Please correct.

Response: We have added the following reference regarding your point:

  • #36. Fukuchi, Y.; Nishimura, M.; Ichinose, M.; Adachi, M.; Nagai, A.; Kuriyama, T.; Takahashi, K.; Nishimura, K.; Ishioka, S.; Aizawa, H.; et al. COPD in Japan: the Nippon COPD Epidemiology study. Respirology 2004, 9, 458-465

  1. Conclusion: It is ok and relevant

Response: We really appreciate your comment.

  1. References: to be reviewed

Response: The following two references have been added to the revision.

  • #13. Stahl, E.; Lindberg, A.; Jansson, S.A.; Ronmark, E.; Svensson, K.; Andersson, F.; Lofdahl, C.G.; Lundback, B. Health-related quality of life is related to COPD disease severity. Health Qual Life Outcomes 2005, 3, 56
  • #36. Fukuchi, Y.; Nishimura, M.; Ichinose, M.; Adachi, M.; Nagai, A.; Kuriyama, T.; Takahashi, K.; Nishimura, K.; Ishioka, S.; Aizawa, H.; et al. COPD in Japan: the Nippon COPD Epidemiology study. Respirology 2004, 9, 458-465.

  1. The language is ok, but there are some sentences not so clear especially in the discussion. I suggest the authors to review the English language with a native English speaker.

Response: Before submitting my revision, I had it edited by a company that specializes in English editing by native English-speaking American researchers.

Reviewer 2 Report

Comments and Suggestions for Authors

The authors have done an excellent job of comparing GOLD and STAR severity classification of COPD in clinical practice. The methodology and statistical tools are well described and appropriate. 

In brief the authors looked at a longitudinal cohort of patients from their center and stratified them based on BODE and START classification and analyzed the disease severity and mortality. 

The results are well summarized and use of Bangdiwala plots is very useful to look at concordance visually and is a validated tool to compare two methods. 

The limitations of the study are well summarized. Small sample, and ethnicity of population are mentioned.

The article adds to the clinical landscape of what is an appropriate tool to classify COPD severity and predict mortality. 

Author Response

Response: Thank you for your careful reading of our manuscript and for your favorable evaluation. We appreciate it very much. We are pleased that many of your comments support our intentions and that we have received a high evaluation. It is a great pleasure to receive such comments, as they are very encouraging for us to carry out our research and publish our paper.